# Development and Evaluation of Amorphous Oral Thin Films Using Solvent-Free Processes: Comparison between 3D Printing and Hot-Melt Extrusion Technologies

**DOI:** 10.3390/pharmaceutics13101613

**Published:** 2021-10-03

**Authors:** Jiaxiang Zhang, Anqi Lu, Rishi Thakkar, Yu Zhang, Mohammed Maniruzzaman

**Affiliations:** Pharmaceutical Engineering and 3D Printing (PharmE3D) Laboratory, Division of Molecular Pharmaceutics and Drug Delivery, College of Pharmacy, The University of Texas at Austin, Austin, TX 78712, USA; jiaxiang.zhang@utexas.edu (J.Z.); anqi.lu@utexas.edu (A.L.); rishithakkar@utexas.edu (R.T.); yu.zhang@utexas.edu (Y.Z.)

**Keywords:** additive manufacturing, amorphous solid dispersion, oral thin film, hot-melt extrusion, dissolution kinetics

## Abstract

Conventional oral dosage forms may not always be optimal especially for those patients suffering from dysphasia or difficulty swallowing. Development of suitable oral thin films (OTFs), therefore, can be an excellent alternative to conventional dosage forms for these patient groups. Hence, the main objective of the current investigation is to develop oral thin film (OTF) formulations using novel solvent-free approaches, including additive manufacturing (AM), hot-melt extrusion, and melt casting. AM, popularly recognized as 3D printing, has been widely utilized for on-demand and personalized formulation development in the pharmaceutical industry. Additionally, in general active pharmaceutical ingredients (APIs) are dissolved or dispersed in polymeric matrices to form amorphous solid dispersions (ASDs). In this study, acetaminophen (APAP) was selected as the model drug, and Klucel™ hydroxypropyl cellulose (HPC) E5 and Soluplus^®^ were used as carrier matrices to form the OTFs. Amorphous OTFs were successfully manufactured by hot-melt extrusion and 3D printing technologies followed by comprehensive studies on the physico-chemical properties of the drug and developed OTFs. Advanced physico-chemical characterizations revealed the presence of amorphous drug in both HME and 3D printed films whereas some crystalline traces were visible in solvent and melt cast films. Moreover, advanced surface analysis conducted by Raman mapping confirmed a more homogenous distribution of amorphous drugs in 3D printed films compared to those prepared by other methods. A series of mathematical models were also used to describe drug release mechanisms from the developed OTFs. Moreover, the in vitro dissolution studies of the 3D printed films demonstrated an improved drug release performance compared to the melt cast or extruded films. This study suggested that HME combined with 3D printing can potentially improve the physical properties of formulations and produce OTFs with preferred qualities such as faster dissolution rate of drugs.

## 1. Introduction

Patients like pediatric, elderly, or those who have difficulty swallowing or dysphasia may often refuse conventional oral dosages such as tablets or capsules objectively [1,2,3]. In addition to the abovementioned populations, there are other patients such as the developmentally disabled, mentally ill, or uncooperative who may be subjectively unwilling to take the conventional oral dosages as well [4,5]. Opening capsules or crushing the tablets could be alternative approaches for such patients to administer conventional oral dosages; however, such approaches might be against the original intention of some specific formulations like coated, controlled released, or multilayer tablets. Even worse, this might not only result in ineffectiveness but also toxicity, and hence was not recommended by the European Medical Agency [6,7]. Thus, it is necessary to develop a dosage form that disintegrates or disperses in the oral cavity such as oral thin film (OTF) which could be optimal for those patients to ease the swallowing problems.

The thin film is usually prepared using water-soluble polymers that can dissolve in the oral cavity, however, the patient adherence is often challenged in such dosage forms, where the patient may swallow the entire or partial film [8,9]. Therefore, the formulations are usually designed where the active pharmaceutical ingredients (APIs) are administrated in the mouth or small intestines [10]. The existence of the polymer might lead to bioadhesive formulation, or the formation of the hydrocolloids once contacted with liquid which allows the drug to be diffused from the film and administered buccally, sublingually, or in the gastrointestinal tracts [11,12,13]. One of the marketed OTFs, ONSOLIS^®^, was designed to dissolve in 30 min after administration [14]. There are approximately 51% and 49% of the total dose absorbed from the buccal mucosa and GI tracts, respectively. Due to the extended dissolution time, film is swallowed with the saliva, then the remaining API gets absorbed in the gastrointestinal tract [15]. The OTF designed to be delivered in the mouth can also bypass the first pass metabolism in the liver and thus improve bioavailability [16,17]. OTFs are emerging as an advanced drug delivery system due to their patient-friendly characteristics, easy manufacturing, accurate dosing, and fast wetting, disintegration, or dissolution [18,19,20].

Several methods that can be used to manufacture the OTFs include solvent casting, melt extrusion, and rolling methods [21,22,23,24]. Solvent casting methods are currently recognized as the most widely used approaches because of the low costs and easy operations [25]. However, the solvent cast films are usually thin (12–100 µm) compared to other preparation methods, leading to potential structural failures during packaging, storage, transportation, or patient handling [26]. Additionally, the use of an organic solvent might raise regulatory concerns or cause additional issues such as environmental pollution or health risks for operators. Melt extrusion is an optimal solvent-free approach for manufacturing the amorphous solid dispersion (ASD) with improved solubility and bioavailability of poorly water-soluble drugs [23,27]. However, the APIs and polymers are exposed to high temperatures and might not be suitable for thermally unstable drugs or excipients.

In recent years, additive manufacturing (AM) has emerged as an attractive technology to fabricate a wide array of pharmaceutical dosage forms. AM, also known as 3-dimensional printing (3D printing), builds objects layer by layer from a computer-aided digital design [28,29]. Extrusion based AM processes, such as fused deposition modeling (FDM), are currently being explored to prepare thin film and membrane formulations [24,30]. However, the film’s design, printing parameters, and the physico-chemical characterization of the printed films have not been fully explored. Additionally, the difference between 3D printed thin films and conventional films has not been thoroughly investigated either. Herein, the current study intends to prove the concept of a synergistic application of melt extrusion with FDM-based AM platforms to manufacture OTFs with robust qualities and faster in vitro release performances.

Three different thermal preparation methods were used to prepare the OTFs in the current investigation: melt casting, hot-melt extrusion, and 3D printing. The primary goals of this study are: (1) to develop acetaminophen (APAP) loaded oral-friendly thin film using solvent-free methods; (2) evaluate and compare different techniques for OTF development with different in vitro techniques; and (3) demonstrate the feasibility of combining AM and HME techniques for personalized or on-demand manufacturing of OTFs.

## 2. Materials and Methods

### 2.1. Materials

Acetaminophen (Sigma–Aldrich, St Louis, MO, USA) was selected as the model drug. The mixture of Klucel^TM^ hydroxypropyl cellulose (HPC) (Ashland Inc. Wilmington, DE, USA) and Soluplus^®^ (BASF Corporation., Florham Park, NJ, USA) was used as the polymeric matrix to form the film. An amount of 30% *w*/*w* of APAP was physically mixed with 50% *w*/*w* of HPC HF grades and 20% *w*/*w* of Soluplus for melt casting or melt extrusion. All the other chemicals including salts, organic solvents, and buffering reagents, were either analytical or HPLC grade.

### 2.2. Preparation of the Oral Dispersible Films

#### 2.2.1. Melt Casting Methods

Physical mixtures were mixed using a mortar and pestle. A CARVER^®^ hydraulic press (Carver, Inc., Wabash, IN, USA) paired with heating components was used to prepare the melt-casted films. The physical mixtures were placed in the middle between the upper and lower plates, which were both set at 160 °C. The force of 10,000 pounds was used to press the films for 3 min. The melt cast films (MCF) were collected and stored in ziplock bags at ambient temperature.

#### 2.2.2. Melt-Extruded Films

The current investigation used a Leistritz 12 mm twin screw corotating extruder (Leistrtz Advanced Technologies Corp., Allendale, NJ, USA) with 8 individually heated barrel zones, where a 1 mm film die was used for forming the film. All extruded films (HME-F) were manually collected. The feeding rate was set at 5 g/min, and the screw speed was set at 75 rpm. The screw configuration and temperature profiles for the barrel zones and die are shown in Figure 1 below.

#### 2.2.3. 3D Printed Films

3D printable filaments were prepared using the identical extrusion setups described in Section 2.2.2., except a 3 mm round-shaped die was used for filament extrusion. As shown in Figure 2 below, the filaments were collected and then subjected to the 3D printing process, where the film was designed using 3D builder software as a rectangular shape with 20, 20, 0.3 mm in L, W, and H, respectively. The 3D models were sliced using Cura software in which line fill patterns and 100% infill densities were selected. In the vertical direction, the films were sliced into 3 layers with each layer thickness of 0.1 mm. A 0.4 mm nozzle was used to build the films, and the printing temperature was set at 170 °C while the printing speed was 50 mm/s. 3D printed films (3DPF) were collected and stored at ambient temperature as well.

### 2.3. Assessment of Films Morphology

Melt cast, melt-extruded, and 3D printed films were cut into 10 × 10 mm pieces, and a Neiko 01407A digital caliper (VWR1, Radnor, PA, USA) was used to determine the thickness and dimensions of the films. The weight of extruded and printed filaments was measured using a Mettler-Toledo ME-TE (Mettler-Toledo, LLC. Columbus, OH, USA) analytical balance. Furthermore, a Dino-Lite AM7391MZTL optical microscope (AnMo Electronics Corporation, Hsinchu, Taiwan) was used to image the samples.

### 2.4. Texture Analysis of the Films

A TA-XT2 analyzer (Texture Technologies Corp, Hamilton, MA, USA) was used to obtain the film burst strength and Young’s module. Films were cut into 20 × 10 mm stripes and mounted with pretension without spacer plate inclusion, A P/4 cylinder probe (diameter = 4 mm) was used to penetrate through the films, where the probe moving speed was set at 1 mm/s for testing, and the stop target was 5 mm after it touched the film. The instrument was operated, and data were collected and analyzed using Exponent Connect software (version 7.0.5.0, Stable Microsystems, Surrey, UK). The Young’s module was estimated via geometry calculations. Six replicates were carried out for each kind of film.

### 2.5. Solid States Analysis

#### 2.5.1. Thermogravimetric Analysis (TGA)

The thermal properties of the raw materials and physical mixtures were determined via a Mettler-Toledo TGA/DSC 1 analyzer (Mettler-Toledo, Schwerzenbach, Switzerland). Pure APAP, HPC, Soluplus, and physical mixtures were placed in an open ceramic crucible, and all samples were ramped from 35 to 400 °C at a rate of 20 °C/min. The furnace was purged using ultra-purified nitrogen at a flow rate of 25 mL/min. The STAR software was used to operate the instrument and collect the data, while data were analyzed using Microsoft Excel software (Version 2007, Microsoft, Redmond, WA, USA).

#### 2.5.2. Differential Scanning Calorimetry (DSC)

MC-F, HME-F, and 3DP-F were collected and cut into small pieces. Raw materials and 5–10 mg of each film sample were placed in the bottom of T-zero aluminum DSC pans, then ramped from 35 to 220 °C at a rate of 10 °C/min. In all DSC experiments, ultra-purified nitrogen was used as the purge gas at a 50 mL/min flow rate. Data were collected and then analyzed using Microsoft Excel Software (Version 2007).

#### 2.5.3. Powder X-ray Diffraction (PXRD)

The solid state of raw materials, physical mixtures, MCF, HMEF, and the 3DPF were investigated via PXRD analysis using a benchtop Rigaku MiniFlex instrument (Rigaku Corporation, Tokyo, Japan). Briefly, all samples were scanned from 5 to 60° 2θ scale, with a scan speed of 2°/min, scan step of 0.02°, and the resultant scan resolution of 0.0025. The voltage was set at 45 V, and the current was set at 15 mA during the scan process. The data were collected and plotted as a stacked plot of 2θ scale versus intensity using Microsoft Excel Software (Version 2007).

#### 2.5.4. Hot-Stage Polarized Light Microscopy (PLM)

The melting behavior of physical mixtures and film crystallinity were analyzed using an Olympus BX53 polarized photomicroscope (Olympus America Inc., Webster, TX, USA) equipped with a Bertrand lens. Physical mixtures were ramped from room temperature to 200 °C at 20 °C/min, while all the film samples were observed at room temperature. A QICAM Fast 1394 digital camera (QImaging, Tucson, AZ, USA) with a 530 nm compensator (U-TP530, Olympus^®^ corporation, Shinjuku City, Tokyo, Japan) was used to capture the images.

### 2.6. Raman Spectroscopy and Raman Mapping

A Nicolet iS50 Raman spectrometer was used to obtain the Raman spectra and Raman images, and the laser was operated at 0.50 W power at the sample. Reference Raman spectra of pure crystalline APAP, HPC, and Soluplus were obtained via scanning from wavenumber 80–4000 cm^−1^. The Raman spectrum of amorphous APAP was determined via melting the APAP and scanned at its molten states. Raman images were taken via scanning 20 × 20 µm area on each sample, where the spectra were collected using the same parameters, and three replicates were scanned for all three kinds of film. Data were collected and analyzed using the OMNIC software (version 9.2.86, Thermo Fisher Scientific Inc., Waltham, MA, USA).

### 2.7. Disintegration Studies and In Vitro Drug Release Study of the Films in Simulated Saliva

Modified disintegration studies were conducted with pictures recorded via the abovementioned Dino-Lite microscope. Samples were placed in a beaker with 30 mL of simulated saliva (SS) (8.00 g/L NaCl, 0.19 g/L KH_2_PO_4_, 2.38 g/L Na_2_HPO_4_, pH = 6.8) [31]. The magnetic stirrer was set at 100 rpm, and picture capture intervals were set at 1 s. Additionally, the drugs released in the SS were also conducted, where 1 mL of samples was withdrawn at time point of 2.5, 5, 10, 15, and 30 min during the disintegration studies.

### 2.8. In Vitro Drug Release Study

The drug release from the films was determined using a United States Pharmacopeia (USP)-II dissolution apparatus (Vankel-Varian VK 7000 dissolution system, Varian, Inc., Cary, NC, USA). Dissolution tests for other formulations were conducted per the US pharmacopeial standards using simulated intestinal fluid^TS^ (USP SIF, without pancreatin) (standard phosphate buffer, 0.02 M KH_2_PO_4_, and NaOH at pH 6.8), which is representative of the small intestinal fluid of humans. Each experiment was carried out in triplicate using 300 mL of the dissolution medium at 37.0 ± 0.5 °C for 24 h. The paddle speed was set at 50 rpm. For analysis, samples were withdrawn at 2.5, 5, 10, 15, 30, 60, and 120 min. The amount of released APAP was determined by HPLC (Agilent 1100 series, Santa Clara, CA, USA) at 243 nm and analyzed using Agilent ChemStation software (version C.01.03, Agilent Technologies, Inc., Santa Clara, CA, USA).

## 3. Results and Discussion

### 3.1. Solid States Analyses

#### 3.1.1. Thermogravimetric Analysis (TGA)

Since a solvent-free thermal process was utilized in the current investigation to achieve molecular level mixing and film forming, a thermal degradation of the drug or excipients used in the formulations might happen due to the high processing temperature of the casting, extrusion, and printing. So, TGA was conducted to assess the thermal behavior of the formulations and select a suitable processing temperature range before performing actual HME and 3D printing [32]. As presented in Figure 3, APAP, polymers, and physical mixtures were chemically stable under 280 °C, where the printing and extrusion temperatures were within such a range. It must be noted that because of the hygroscopicity and due to the moisture content, there was around 3% of weight loss presented in the TGA curves of the bulk polymers and physical mixtures. Apart from that, there is no additional potential degradation of the drug or excipients observed as a function of weight loss in any of the TGA traces. Thus, the TGA results confirmed that there is likely no thermal degradation during the thermal processing.

#### 3.1.2. Differential Scanning Calorimetry (DSC)

As shown in Figure 4, the APAP exhibited a sharp endothermic peak with an onset at 172.7 °C corresponding to its melting transition, whereas both Soluplus and HPC exhibited glass transitions at around 78 and 130 °C, respectively. This may indicate that both polymers are likely to soften at or above 160 °C [33]. The physical mixtures presented an attenuated melting peak at around 149.77 °C, which is primarily because of the interaction between the APAP and polymer molecules, such as formation of the non-covalent bonds. Such melting behavior can also be confirmed by PLM observations which has been discussed in detail in Section 3.1.4.

The APAP was expected to dissolve or be distributed into the molten polymeric matrix and form amorphous solid dispersions during the thermal process used in this work. As demonstrated in Figure 4, the HME-F and 3DP-F showed no pronounced endothermic peak during the heating process, indicating the formation of amorphous film after the extrusion process, while there was an attenuated peak at 159.35 °C in MC-F, which indicated the existence of crystalline APAP. Further PXRD characterization was conducted to confirm the crystallinity of each extruded or printed film.

#### 3.1.3. Powder X-ray Diffraction Analysis of the Crystallinity

The PXRD analysis was carried out to identify the crystalline APAP and the existence of crystalline APAP in the melt cast, extruded, and 3D printed films. PXRD is one of the most widely used approaches to differentiate the crystalline and amorphous solids because of the long-range molecular order in response to the X-ray diffractions [32].

As shown in Figure 5, pure APAP exhibited characteristic peaks at around 2-theta of 11.94, 15.44, 24.20, and 36.78 degrees, which confirmed its crystalline characteristics. However, both Soluplus and HPC HF grades lack long-range molecular order, leading to the lack of Bragg’s peaks observed from the powder X-ray absorption.

As demonstrated in Figure 5, the MC-F showed attenuated peaks compared to the raw APAP, which confirmed the existence of the crystalline APAP in the film. Unlike the MC-F, both the HME-F and 3DP-F showed no crystalline peaks corresponding to the drug, which confirmed the amorphization of APAP. The PXRD and DSC results can cross verify each other and confirmed that the process condition of HME allows for the formation of ASDs while melt casting does not.

#### 3.1.4. Hot-Staged Polarized Microscopy (PLM)

A hot stage conjugated with polarized microscope can be used to confirm the melting behavior of the materials visually. As demonstrated in Figure 6a, the smaller needle-shaped crystals were APAP, and the prismatic translucent particles were HPC, while the round-shaped opaque particles were Soluplus. The polymers possessed a lower glass transition temperature than the melting point of APAP, and some particles started melting at around 120 °C. The APAP slowly dissolved into the molten polymeric matrix during the raising of the temperatures and completely melted at around 167 °C. Complete melting before the melting points can also be confirmed by DSC studies, where there is an attenuated endothermic peak at around 149.77 °C. Additionally, when the molten polymer was cooling down to room temperatures, no APAP crystal could be observed under the microscope which potentially indicates the formation of the ASD.

As shown in Figure 6b, the 3DP-F were more uniform and the drug distributed more evenly, while MC-F and HME-F were not as uniform as 3DP-F. Regarding MC-F, the drug was observed embedded in the transparent polymers at higher magnification (bottom-right corner picture in Figure 6b), leading to drug enrichment in a specific area of the MC-F (top-right corner picture in Figure 6b). Such local enrichment of the drug in MC-F is mainly due to the lack of mixing effect during the casting process. In other words, the drug was distributed more uniformly in HME-F because of the high shear mixing and kneading during extrusion. Moreover, the FDM printing process melts the filaments once again, leading to molecular level mixing and a more even distribution of the drug in 3DP-F than HME-F.

### 3.2. Raman Analysis

As demonstrated in Figure 7, the Raman spectra of amorphous and crystalline APAP, HPC, and Soluplus matched the previous reports [34,35,36], where the crystalline APAP demonstrated a sharp peak at around 1648 cm^−1^ indicating the C=O stretching movements while the amorphous APAP showed an attenuated and broad band as well as upshifted peak at around 1656 cm^−1^. Additionally, there were no noticeable peaks observed within the range of 80–1800 cm^−1^ in the HPC and Soluplus spectra.

Therefore, the 1648 cm^−1^ peak was used to represent the crystalline APAP, while the 1656 cm^−1^ peak was used to represent the amorphous APAP during the Raman imaging/mapping analysis. The Raman intensity maps of crystalline and amorphous APAP are shown in Figure 8a,b, where the intensity strength was represented as the rainbow-colored intensity scale. The higher the intensity, the more the match to the reference crystalline or amorphous APAP spectrum. In order to directly compare Raman maps, an intensity scale of 0–3.0 was used in the current investigation.

As demonstrated in Figure 8a, the Raman intensity of 1648 cm^−1^ represented the crystalline APAP within each sample. The map of 3DP-F and HME-F showed almost no red color in the Raman map, indicating the minute quantity of crystalline APAP in each kind of film. On the contrary, the map of MC-F showed considerable amounts of red color, which indicates that there are still some crystalline APAP. Furthermore, such Raman mapping observations can be cross verified via DSC and PXRD results where the small amounts of crystalline APAP were detected in the MC-F. The Raman maps of intensity at 1656 cm^−1^ are demonstrated in Figure 8b, where the amorphous APAP can be identified in all three kinds of films. The intensity of the amorphous APAP in 3DP-F and HME-F (Figure 8b) was more substantial than the MC-F and combined the abovementioned minute quantity of crystalline APAP in Figure 8a that indicates the amorphorization of APAP during the HME and 3D printing process. Additionally, as observed in Figure 8a,b, the rainbow colors were distributed more uniformly in 3DP-F and HME-F, while APAP were observed concentrated in a small area of MC-F. This observation indicates that the drug was distributed more uniformly in the 3DP-F and HME-F.

#### 3.2.1. Impact of the Preparation Methods on the Films

HPC and Soluplus have been used in pharmaceutical product development as film formers which offer a smooth and neat appearance for the film. The different preparation methods used in the current investigation could be the critical variable that significantly affects the crystallinity, physico-chemical property, and quality of the ODFs. The comparision studies between three different kinds of films are discussed thoroughly in the following sections.

#### 3.2.2. Appearance of the Films

As shown in Figure 9, the smoothness of the film is observed in the order of MC-F > HME-F > 3DP-F. The lack of mixing and friction during the casting process will result in the smooth surface of the MC-F. The 3DP-F possessed the roughest surface among three kinds of films because of the layer-by-layer building mechanisms of the FDM printer, where the nozzle was rubbing on the newly deposited layer.

The melt cast film also exhibited a transparent-looking form while crystalline APAP can be confirmed by the naked eye, indicating the current process condition (10,000 pounds at 160 °C for 3 min) can not allow all the APAP to dissolve into the molten polymeric matrix. The existence of the crystalline APAP can also be confirmed by the abovementioned solid states analysis (discussed in Section 3.1).

The melt-extruded film also exhibited a transparent strip-shaped morphology, and the extrusion trajectory can be clearly observed. In addition, the HME-F presents fewer amounts of the crystalline APAP than the MC-F, mainly because of the kneading and mixing during the extrusion process that helps the APAP dissolve or disperse into the molten polymer matrix.

The 3D printed film was opaque and showed a rectified shape, mainly because the film morphology can be manipulated by the 3D design. However, it is hard to identify the crystalline ingredients based on the appearance of the film. As shown in the polarized microscopies, APAP was distributed more uniformly in 3DP-F compared to the HME-F and MC-F.

#### 3.2.3. Amorphization of the APAP

One of the primary goals of the current investigation is to develop the amorphous oral dispersible film via solvent-free methods. As discussed in Section 3.1, the crystalline APAP in the films indicates the crystallinity of the final film products, and it also reveals advancements in three different thermal processes. There are considerable amounts of crystalline APAP observed in the MC-F, which indicates that the heating, residence time, and pressure during the casting process have limited attributes to API’s complete amorphization. Even though the polymer matrix was melted at 165 °C and might allow APAP to dissolve or distribute into the matrix, the pressure had no mixing effects and time was not long enough to allow all the APAP to be amorphized or distributed into the matrix at the molecular level.

On the other hand, with the help of the kneading and mixing attributes from the extrusion process, all the crystalline APAP was completely converted to its amorphous counterpart and formed ASDs (HME-F and filaments for 3D printing). Even though the extrusion temperature was the same as the melt casting, the extrusion process will accelerate the dissolution or distribution of the APAP into the molten polymers via mixing. Additionally, the complete amorphization of the APAP in the filaments also leads to the amorphous 3DP-F.

#### 3.2.4. Process Quality

Another critical purpose of this work is to study the different thermal processes of preparing oral films. All the collected films were cut into 1 mm × 1 mm square-shaped samples, in which the dimension and thickness were measured, and the densities were estimated. As demonstrated in Table 1, the dimension variations of the MC-F and HME-F were relatively more extensive compared to the 3D printed films. The melt casting of the film could be considered as a batch process where the batch-to-batch variation is not negligible. Additionally, the uniformity of the HME-F was dominated by the pulling process, where the thickness of the films can be manipulated via adjusting the pulling speeds. 3D printing is a stable process that can produce products with consistent qualities. Even though the 3D printing process is also a batch process, the reproducibility is better than casting and extrusion, especially the thickness of the 3DP-F presented 0.00 variations. It must be noted that the measured thickness of the 3DP-F was smaller than the designed thickness, 0.3 mm, which is mainly due to the condensation of molten materials after cooling down.

### 3.3. Texture Analysis

The mechanical properties of the ODFs should be evaluated in order to predict the potential structure failure during storage, transportation, or on the market [37]. That is to say, the films should have adequate strength (burst force), where the deformations occurred before the film burst should be elastic deformation for OTFs prepared via different methods. In this work, a film burst test that offered breaking force changed as a function of the distance of penetrating probe travel was conducted to determine the film strength and the modules.

As shown in Figure 10a, the breaking force (*F*) is determined as the largest force during the texture analysis, while the breaking distance (*D*) is the distance when reaching the highest force. Based on the observation, the deformation of the film before reaching the breaking force shall be elastic deformation, and the new length of the film changed before burst was recorded as the *L_n_* and calculated using the Pythagorean Equation (1) shown below:(1)$$L_{n}=2\times \sqrt{{\left(\frac{1}{2}\times L_{0}\right)}^{2}+D^{2}}$$
where *L*_0_ is the length of the original film being mounted between the gap, so it is equal to the gap of the testing rig (Figure 10a). The Young’s module was estimated using the Equation (2) as below [38]:(2)$$Young^{\prime }s\;module,\;E=\frac{Stress}{Starin\;}=\frac{F/A}{\Delta L/L_{0}}$$
where *F* is denoted as the breaking force, and *A* is the surface area of the 4 mm cylinder-shaped probe (≈12.57 mm^2^). The ΔL is the length of the film changed, which should be the difference between the new length *L_n_* and the original length *L*_0_.

As demonstrated in Figure 10b (limited by the clarity, only one of each kind of film was demonstrated), all three kinds of film showed similar deformation behaviors, where the force builds up near-linear before burst and a sharp drop of force was observed at burst point. As shown in Figure 10c, the 3DP-F showed relatively larger breaking force and distances than the other films, indicating the adequate mechanical properties of 3D printing films. Additionally, the lower burst force and distance of MC-F may lead to potential structural failures after manufacturing. In fact, the HME process is a typical approach to strengthening and toughening polymeric materials due to the mixing and kneading effects during the melt extrusion process, and it can be confirmed that the mechanical properties of HME-F were relatively better than the MC-F. Additionally, due to the layered structure and the orthogonal cross-linked layer, 3DP-F showed even higher breaking force and higher elastic module compared to the two other films.

### 3.4. Film Disintegration

The disintegration time of thin film formulations is vital because it pre-dominates the onset of drug dissolution or delivery. The disintegration time of most of the fast dissolving thin films is within 30 s, while the films prepared in the current investigation are designed to be administrated in the mouth cavity or enteric. As shown in Figure 11, the thin films demonstrated extended disintegration times of 12.8–21.5 min, which was dominated by the densities or thickness of the film and the nature of the control released polymeric matrix.

There are two scenarios that were assumed during the 3DP-F disintegration studies:

(1).The rate of forming hydrocolloids is faster than the rate of dissolution or erosion. The film will completely form hydrocolloids where diffusion dominated the drug release, then the erosion or the dissolution of the polymeric matrix dominates the drug release.(2).The rate of forming hydrocolloids is slower than or equal to the rate of dissolution or erosion. Both the diffusion and erosion/dissolution will dominate the drug release till the end.

In practice, the disintegration of MC-F and HME-F showed similar behaviors and can be considered as case 1), where the film slowly formed hydrocolloids while drug diffused from the colloids to the media, then erode or dissolve to release the drugs. At the early stages (first two photos of MC-F and HME-F shown in Figure 11), the films still shown robust structure, where the entire film rotates with stirring (<10 and 15 min for MC-F and HME-F, respectively). In this stage, the media interacts with the polymeric matrix and forms hydrocolloids, while the drug within the intermediate layers will slowly diffuse into the media due to the concentration gradient. After a certain period, the entire film was formed as hydrocolloids where the films were becoming soft and easy to tear (the third photos of MC-F and HME-F shown in Figure 11). Due to the softening of the objects, the film can be torn apart by the stirring, where small pieces are shown in the dissolution media (last photos of MC-F and HME-F shown in Figure 11). In the current investigation, the time point of small pieces that can be seen is defined as the disintegration time for the OTFs.

Due to the 3D design, the 3DP-F showed different dissolution or disintegration behaviors compared to the other two films. The 3DP-F will also form hydrocolloids when contacted with the media. However, the rate is slower than the rates of the other two films.

### 3.5. Drug Release from the Films

#### 3.5.1. Dissolution in Simulated Saliva

As observed from the dissolution studies in 30 mL of SS, the MC-F and HME-F were floating on the surface of the dissolution media, which might be because of their lower densities (Table 1). After thoroughly wetting or forming the hydrocolloids with aqueous media, the MC-F and HME-F were sinking from the surface to the bottom. The drug release profiles of MC-F and HME-F were almost similar because of their identical physical properties. Before 5 min, MC-F released the drug faster which could be attributed to its faster disintegration compared to the HME-F. Moreover, HME-F released the drug faster after 5 min which could be due to the formation of the ASD which lowered the glass transition temperature of the entire film system and resulted in faster interactions with the aqueous medium for faster erosion or dissolution. On the contrary, the 3DP-F rapidly sank into the bottom after wetting because its density was relatively heavier than water. As demonstrated in Figure 12a, the 3DP-F showed faster drug release rates because of the rapid sinking and reached around 32.2% and 56.5% of APAP released in 5 min and 30 min, respectively.

Several kinetic models have been widely used by pharmaceutical scientists to understand release kinetics or mechanisms during formulation development comprehensively. As mentioned above, the film will form hydrocolloids because of the hydrophilic polymers, where the release of the APAP from the film was dominated by the diffusion, swell, or erosion of the films. First-order, Higuchi [39], Korsmeyer-Peppas [40], and Sahlin-Peppas [41] models were applied to describe in vitro drug release kinetics in the current investigation.

A first-order model can be described as the drug release following the first-order kinetics where the release rate is proportional to the drug concentration and can be expressed as follows [42]:(3)$$\frac{dC}{dt}=k_{1}\times C$$
where *k*_1_ is the rate constant, while *C* is the concentration of the drug, and t is the time.

The Higuchi model was firstly developed in 1961 by Higuchi [43,44], and the model was expressed as follows:(4)$$\frac{M_{t}}{A}=\sqrt{D\left(2\times C_{0}-C_{s}\right)\times C_{s}\times t}$$
where *Mt* is the drug released at time *t* and *A* is the contact area of the medium and dosages; *D* is the drug diffusivity in the polymer; *C*_0_ is the initial drug concentration, while *C_s_* is the solubility of the drug in media. Such an equation can be simplified as follows [39]:(5)$$\frac{M_{t}}{M_{\infty }}=k_{H}\times t^{1/2}$$
where *k_H_* is the rate constant closely related to the structure or geometry of the dosages.

The Korsmeyer–Peppas model, also popularly recognized as the Ritger–Peppas model, or power law, was developed by Peppas et al. in 1983 [45,46], which can exponentially explain drug release kinetics and can be expressed as follows:(6)$$\frac{M_{t}}{M_{\infty }}=k\times t^{n}$$
where *M_t_* and *M*_∞_ are drugs released at time *t* and the total amount of drug, *k* is the rate constant, *t* is the time, and *n* is the release exponent closely related to the mechanism of drug release.

The Peppas–Sahlin model also developed a model to analyze anomalous transportation [47]:(7)$$\frac{M_{t}}{M_{\infty }}=k_{1}\times t^{m}+k_{2}\times t^{2m}$$
where $k_{1}$, $k_{2}$, and m are constants. The first term on the right-hand side represents the Fickian diffusional attributes, while the second term has non-Fickian attributes.

The rate constants, release exponents, and the R-square of the abovementioned four models are listed in Table 2. The release is not following the first-order release or Higuchi model and can be confirmed by the poor fitting/*R*^2^. The Higuchi model is only applicable for “ideal” controlled released systems, and the use of such a model was limited by assumptions including (1) *C*_0_ >> *Cs*; (2) edge effects are negligible (diffusion is one-dimensional); (3) swelling or erosion of the polymeric matrix is negligible; (4) sinker condition must be maintained [39,48]. The drug release from all three kinds of films might not meet all the assumptions, which indicates the swell or dissolution of the polymer should not be neglected, and also confirmed the observation of disintegration studies and the drug release profiles.

The drug release profiles showed better fitting in the Korsmeyer–Peppas model, where the *R*^2^ is relatively greater than the first-order or Higuchi fittings. Additionally, there are two distinct values of the release exponent, *n* has to be noticed: (1) if *n* < 0.45 indicates the drug release is dominated by Fickian diffusion; (2) while *n* > 0.89 means swell or dissolution of the matrix is dominating the drug release [46,49,50]. As listed in Table 2, the drug release from the MC-F and 3DP-F were entirely dominated by Fickian diffusion, while the HME-F were anomalous kinetics (dominated by both diffusions and swell).

The Peppas–Sahlin model showed adequate confident intervals (*R*^2^ > 0.9560) of all three different kinds of films, which indicates the Fickian diffusion kinetics take the lead during the drug release from the OTFs. As calculated from this model, the *k*_1_ > *k*_2_, and k_2_ is negative for all OTFs, which indicated that the Fickian diffusion is the primary factor that dominates the drug release rather than the relaxation effects. Additionally, the Peppas–Sahlin model was further developed to determine the factor of the Fickian drug release mechanism, F, which can be calculated as [40,41]:(8)$$F=\frac{1}{1+\frac{k_{2}}{k_{1}}t^{m}}$$

As shown in Figure 12b, factor *F* is relatively small, and the drug release might be dominated by anomalous kinetics where the polymeric chain and diffusion both affect the release. After completely forming the hydrocolloids, the factor F grew exponentially, and the drug was diffused from the polymeric matrix to the liquid-colloid interface then released to the media, where the diffusion mechanisms completely dominated such behavior. Additionally, as shown in Figure 12b, the factor F of amorphous HME-F and 3DP-F was relatively larger than that of MC-F, which indicates the amorphous film formed hydrocolloids faster and released the drug more constantly compared to the non-ASDs.

#### 3.5.2. Dissolution in SIF

Considering the patient compliance or adherence, the following situations may happen to the patients after the administration of the OTFs: (i) the drug might dissolve and be completely absorbed in oral mucosa or (ii) the film could partially dissolve in the mucosa, and other parts swallowed with saliva to the GI tract and then eventually absorbed in the intestine, and (iii) the patient could swallow the whole film accidently, which would make the drug dissolve and get absorbed mainly in the intestine. During the dissolution studies in the 300 mL of SIF, all the samples were floating on the surface of the media initially and then slowly sank. In this case, the drug release from 3DP-F showed faster drug release compared to that in the SS, where 3DP-F reached 60.42% drug released in SIF (Figure 13a) while 56.54% in SS at 30 min. Since the sink condition was maintained during both dissolution studies, such a difference might cause the floating in SIF that slows down the wetting or sinking and leads to smaller contact area with media. Additionally, HME-F almost maintains the same drug release profiles, where they reach 30.74% and 29.04% of APAP at 30 min in SIF and SS, respectively. Since a similar buffer system was used as SIF or SS and pH was maintained the same, the type of ions will not significantly affect the drug release behaviors. It has to be noted that the drug release from MC-F in SIF was slightly slower than released in SS, where the MC-F reached 24.56% drug released in SIF while 27.79% in SS at 30 min. Moreover, as demonstrated in Figure 13b, the factor F of amorphous 3DP-F was relatively larger than that of HME-F and MC-F, which indicates the amorphous film formed hydrocolloids faster and released the drug more constantly compared to the non-ASDs. Additionally, the factor F trends were observed to be identical in HME-F and MC-F (varies from the dissolution in the SS), which might be affected by the stirring, environment temperature, or other errors.

In addition, in order to comprehensively understand the release kinetics, the profiles were also fitted into the abovementioned mathematical models. The fitting parameters and *R*^2^ are listed in Table 3.

As expected, the Peppas–Sahlin model offered the highest *R*^2^ so that a similar conclusion can be drawn, where the diffusion mechanisms dominate the drug release from OTFs in SIF.

To sum up the in vitro dissolution studies, the drug release from SS and SIF was similar, indicating that the drug can be absorbed both from the mouth cavity and intestine. The release mechanism of the OTFs first interacts with the media followed by forming hydrocolloids, and then the diffusion mechanisms will dominate the drug release from the matrix. Moreover, the formation of the amorphous solid dispersion will result in the faster formation of the hydrocolloids, which can accelerate the drug release from HME-F and 3DP-F compared to the non-amorphous film, MC-F.

## 4. Conclusions

In the current investigation, three different kinds of films were successfully fabricated via solvent-free methods. Additionally, the amorphous solid dispersions were successfully prepared via HME and 3D printing technologies, which improved the drug release rate from the hydrophilic polymeric matrices. Filaments prepared using binary polymer blends of HPC and Soluplus were suitable for FDM-based 3D printing. The physico-chemical properties of films were characterized, and comparison studies were carried out between MC-F, HME-F, and 3DP-F. The 3DP-F exhibited a more consistent and elegant appearance, more uniform drug distribution, as well as improved in vitro drug release performance.

Notably, a comprehensive study of the OTFs preparation using 3D printing techniques demonstrated that such film formulations could help patients with difficulty swallowing. In addition, this work also offered a future perspective on on-demand manufacturing of personalized drug delivery systems using pharmaceutical additive manufacturing approaches.

## Figures and Tables

**Figure 1 pharmaceutics-13-01613-f001:**
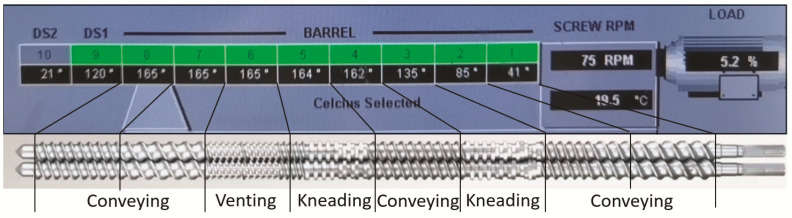
The screw configuration and temperature profiles of the extrusion process.

**Figure 2 pharmaceutics-13-01613-f002:**
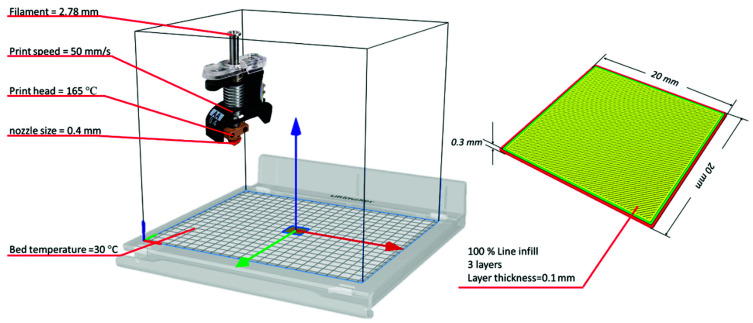
The printing process parameters and the film 3D design.

**Figure 3 pharmaceutics-13-01613-f003:**
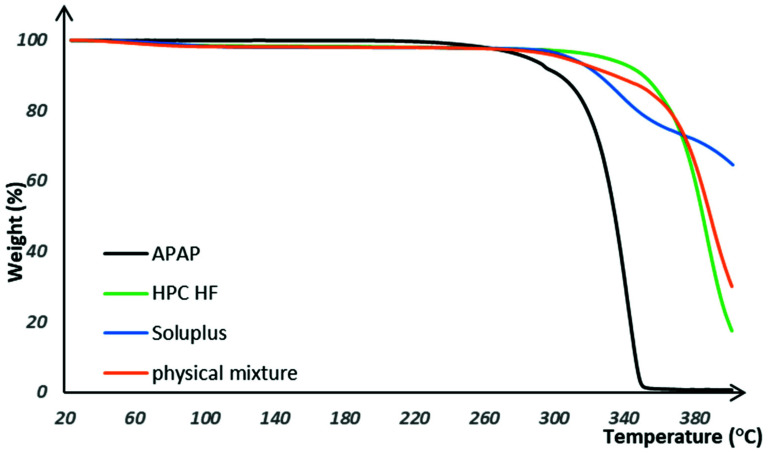
The thermal degradation curves for raw materials and physical mixtures.

**Figure 4 pharmaceutics-13-01613-f004:**
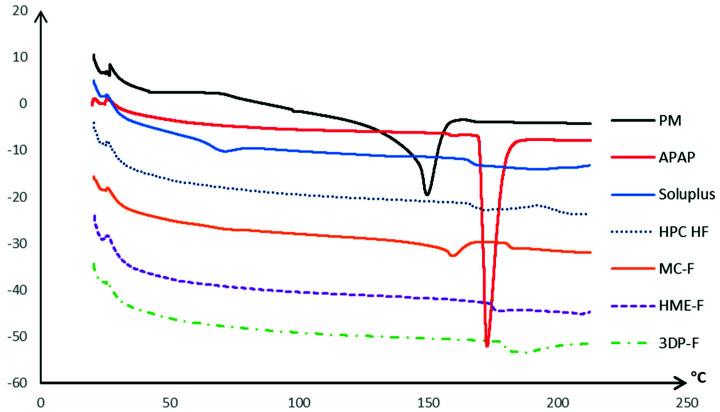
The DSC curves of the raw materials, MC-F, HME-F, and 3DP-F.

**Figure 5 pharmaceutics-13-01613-f005:**
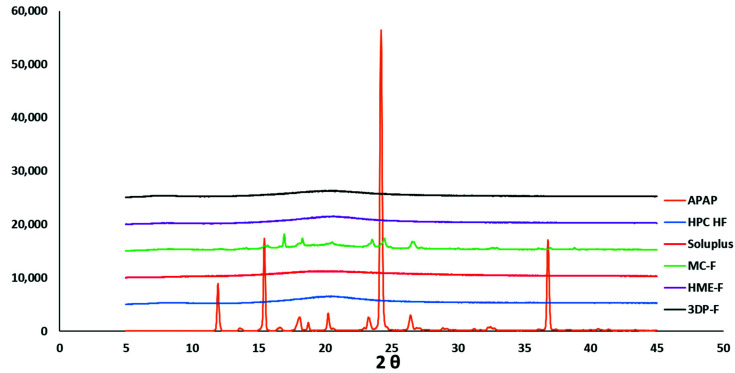
P-XRD curves of raw materials and films prepared via different methods.

**Figure 6 pharmaceutics-13-01613-f006:**
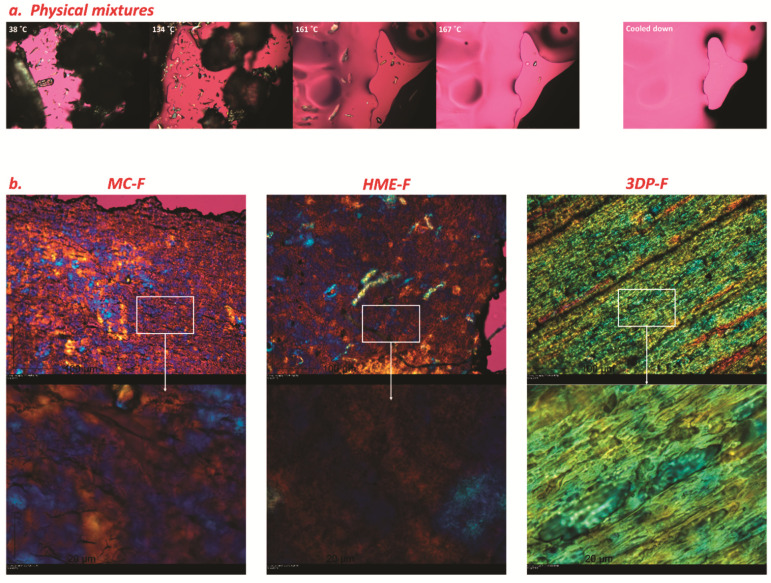
(**a**) The observation of physical mixtures ramped from room temperature to 180 °C, then cooled down to room temperature; and (**b**) morphology of different kinds of films under room temperature.

**Figure 7 pharmaceutics-13-01613-f007:**
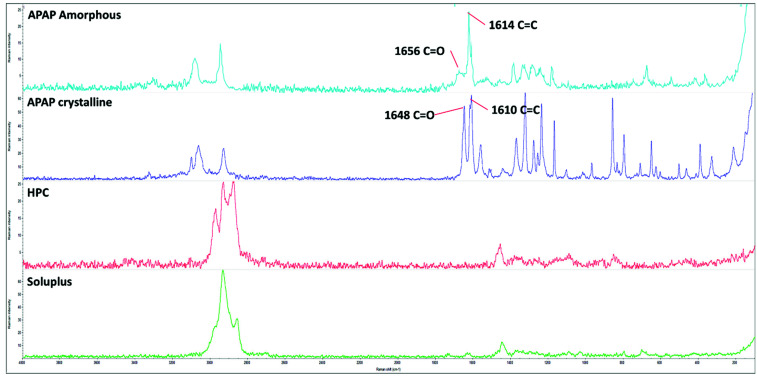
Raman spectra of the amorphous APAP (light blue), crystalline APAP (dark blue), HPC (red), and Soluplus (green) polymer.

**Figure 8 pharmaceutics-13-01613-f008:**
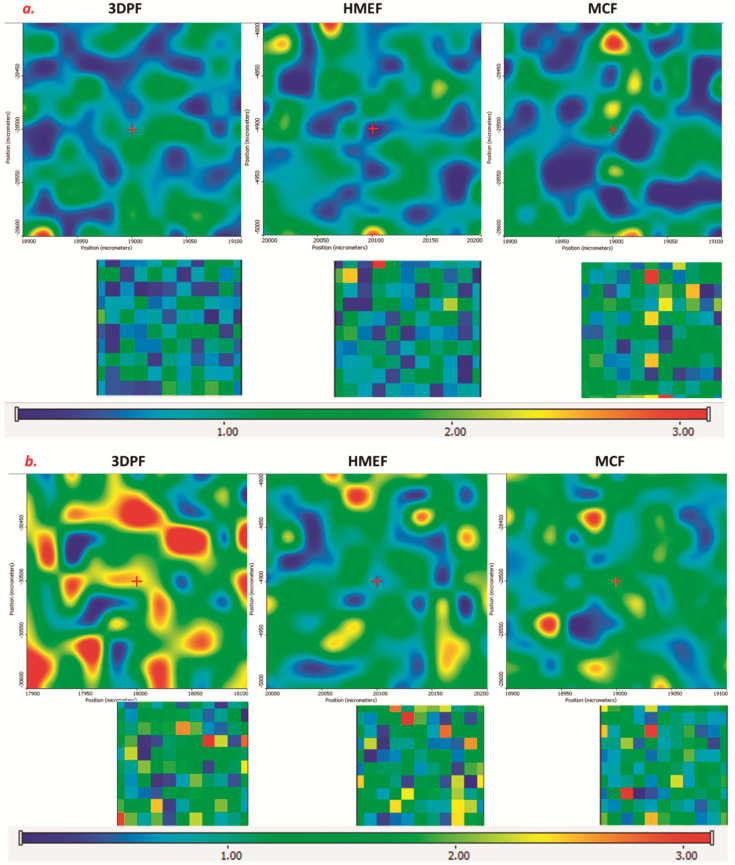
(**a**) Raman maps of the 3DP-F, HME-F, and MC-F at the Raman shift of 1648 cm^−1^; and (**b**) 1656 cm^−1^.

**Figure 9 pharmaceutics-13-01613-f009:**
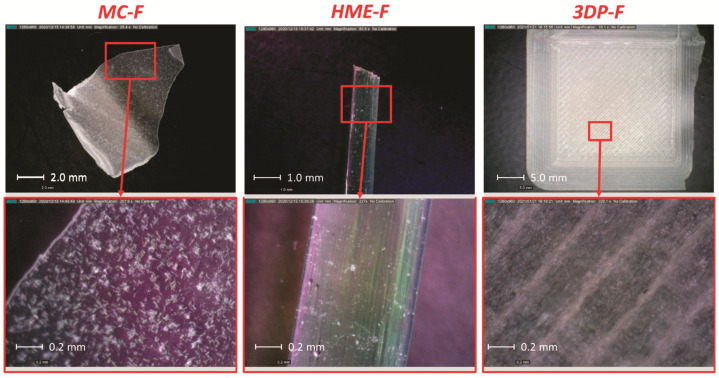
The demonstration of the MC-F, HME-F, 3DP-F under the optical microscope and polarized light microscope.

**Figure 10 pharmaceutics-13-01613-f010:**
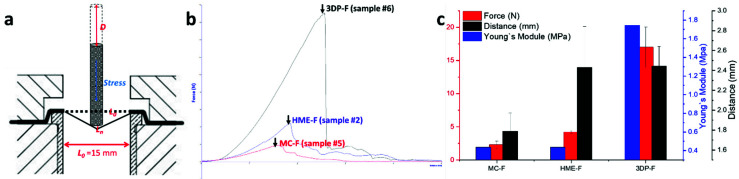
(**a**) Demonstration of the film burst test setups; (**b**) force-distance curve of the film burst test; (**c**) the breaking force, distance, and Young’s module of MC-F, HME-F, and 3DP-F.

**Figure 11 pharmaceutics-13-01613-f011:**
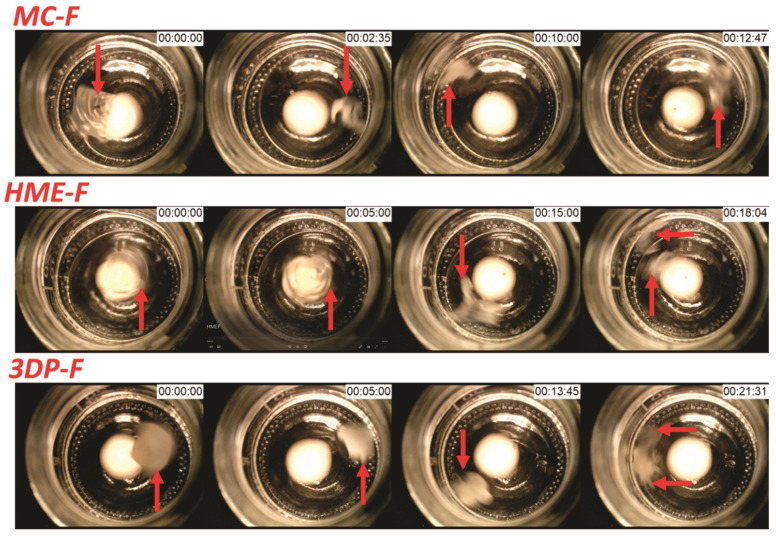
Demonstration of the modified disintegration studies for MC-F, HME-F, and 3DP-F.

**Figure 12 pharmaceutics-13-01613-f012:**
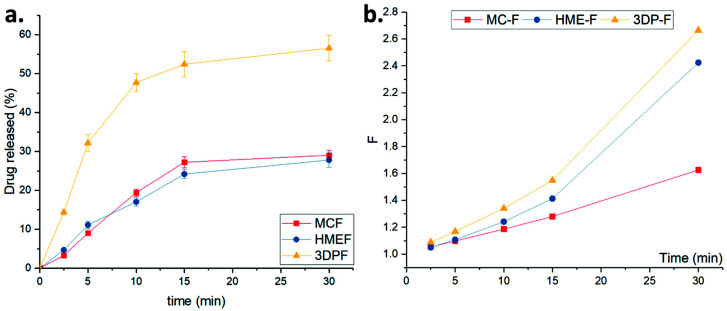
(**a**) The drug release profiles and (**b**) the Fickian diffusion factor, F, of the MC-F, HME-F, and 3DP-F in SS.

**Figure 13 pharmaceutics-13-01613-f013:**
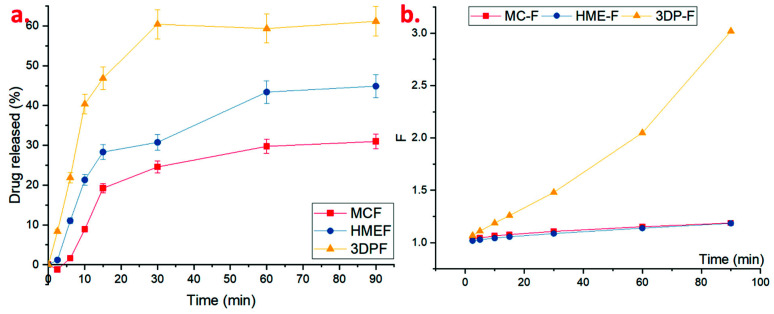
(**a**) The drug release profiles and (**b**) the Fickian diffusion factor, F, of the MC-F, HME-F, and 3DP-F in SIF.

**Table 1 pharmaceutics-13-01613-t001:** The dimensions, weight, and calculated densities of the MC-F, HME-F, and 3DP-F. (*n* = 9, arithmetic mean ± SD).

Formulation	MC-F			HME-F			3DP-F		
	W (mg)	T (mm)	ρ (kg/m^3^)	W (mg)	T (mm)	ρ (kg/m^3^)	W (mg)	T (mm)	ρ (kg/m^3^)
Average	17.33	0.21	0.84	19.33	0.24	0.92	27.67	0.26	1.06
SD.	2.08	0.03	0.02	4.93	0.06	0.03	0.58	0.00	0.02
RSD %	12.01	12.18	1.94	25.51	26.10	2.91	2.09	0.00	2.09

**Table 2 pharmaceutics-13-01613-t002:** The model fitting parameters of the dissolution study in SS.

Films	First-Order		Higuchi		Korsmeyer–Peppas			Peppas–Sahlin			
	*k* _1_	*R* ^2^	*k_H_*	*R* ^2^	*k_KP_*	*n*	*R* ^2^	*k* _1_	*k* _2_	*m*	*R* ^2^
MC-F	0.009	0.4642	4.766	0.8216	6.850	0.394	0.8504	3.233	−0.077	0.817	0.9560
HME-F	0.014	0.8217	5.313	0.9510	4.733	0.541	0.9536	2.196	−0.042	1.007	0.9958
3DP-F	0.046	0.7650	12.133	0.8977	16.607	0.388	0.9243	9.243	−0.356	0.819	0.9880

**Table 3 pharmaceutics-13-01613-t003:** The model fitting parameters of the dissolution study in SIF.

Films	First-Order		Higuchi		Korsmeyer–Peppas			Peppas–Sahlin			
	*k* _1_	*R* ^2^	*k_H_*	*R* ^2^	*k_KP_*	*n*	*R* ^2^	*k* _1_	*k* _2_	*m*	*R* ^2^
MC-F	0.024	0.5600	8.182	0.7610	16.383	0.319	0.8701	9.062	−0.311	0.660	0.9798
HME-F	0.006	0.7584	3.586	0.8765	3.101	0.537	0.8789	1.289	−0.013	0.928	0.9740
3DP-F	0.009	0.7176	5.322	0.9180	6.543	0.447	0.9245	3.779	−0.079	0.731	0.9803

## Data Availability

The data presented in this study are available on request.

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
