# Peer review of "Development and Evaluation of Amorphous Oral Thin Films Using Solvent-Free Processes: Comparison between 3D Printing and Hot-Melt Extrusion Technologies"

_pharmaceutics, 2021, doi:10.3390/pharmaceutics13101613_

Round 1

Reviewer 1 Report

The present paper deals about the development of oral thin film for buccal administration of drugs. The molecule acetaminophenon was used as model. In particualr the study was focused on the evaluation of three different methods for thin film production: 3D printing, hot melt extrusion and casting. Despite the topic is interesting in my opinion some aspects of the study are not clear and must be revised.

Par. 2.7. Line 184. Please indicate the reference for the simulated salivary fluid composition.

Par. 2.8. In vitro release studies. In the introduction authors state “The OTF designed to be delivered in the mouth can also bypass the first pass metabolism in the liver and thus improve bioavailability” (lines 52-53). Base on this statement the formulation should be applied in the mouth and the drug released and absorbed in the oral mucosa. Why the authors performed the release studies in simulated intestinal fluid?? (lines 192-193).

Line 452. Dissolution in simulated saliva. In method section it is not described the use of this medium for dissolution studies. Please clarify.

If the formulation is projected to be applied on buccal mucosa by bioadhesion, the experiment must be planne din order to simulate the application conditions (see e.g. the following publication: Preformulation studies of mucoadhesive tablets for carbamazepine sublingual administration, Colloids and Surfaces B: Biointerfaces 102 (2013) 915 – 922).

Authors state: Lines 49-51. “The existence of the polymer might lead to bioadhesive formulation, or the formation of the hydrocolloid once contacted with liquid and allows the drug to be diffused from the film and administered buccally or sublingually”. As in the composition was used HPC, why bioadhesion studies were not performed?

There are many errors. Please check the english and formatting.

Author Response

Reviewer 1

The present paper deals about the development of oral thin film for buccal administration of drugs. The molecule acetaminophen was used as model. In particular the study was focused on the evaluation of three different methods for thin film production: 3D printing, hot melt extrusion and casting. Despite the topic is interesting in my opinion some aspects of the study are not clear and must be revised.

  1. 2.7. Line 184. Please indicate the reference for the simulated salivary fluid composition.

Reference was added and highlighted.

  1. 2.8. In vitro release studies. In the introduction authors state “The OTF designed to be delivered in the mouth can also bypass the first pass metabolism in the liver and thus improve bioavailability” (lines 52-53). Base on this statement the formulation should be applied in the mouth and the drug released and absorbed in the oral mucosa. Why the authors performed the release studies in simulated intestinal fluid?? (lines 192-193).

Thanks for pointing out the above concern on drug release studies. In fact, the authors attempted to consider all potential scenarios that may happen to the patients after the administration of the films: (1) the drug would  dissolve and completely get absorb in oral mucosa or (2) the film could partially dissolve in the mucosa, and other parts get swallowed with saliva to the GI tract and then eventually gets absorbed in the intestine, and (3) the patient could swallow the whole film accidently, which would make the drug dissolve and get absorbed mainly in the intestine.

For example, one of the marketed Fentanyl OTF, ONSOLIS, has “approximately 51% of the total dose absorbed from the buccal mucosa, and the remaining 49% of the total dose is swallowed with the saliva and then slowly absorbed from the GI tract.”(please refer to: https://www.accessdata.fda.gov/drugsatfda_docs/label/2016/022266s017s018lbl.pdf)[1]

So, by considering the above potential scenarios the in vitro drug release studies were performed in two conditions.

  1. Line 452. Dissolution in simulated saliva. In method section it is not described the use of this medium for dissolution studies. Please clarify.

Thanks for pointing out the issue. In fact, the disintegration is also the drug release in SS, section was modified and highlighted.

  1. If the formulation is projected to be applied on buccal mucosa by bioadhesion, the experiment must be planned in order to simulate the application conditions (see e.g. the following publication: Preformulation studies of mucoadhesive tablets for carbamazepine sublingual administration, Colloids and Surfaces B: Biointerfaces 102 (2013) 915 – 922).

Authors state: Lines 49-51. “The existence of the polymer might lead to bioadhesive formulation, or the formation of the hydrocolloid once contacted with liquid and allows the drug to be diffused from the film and administered buccally or sublingually”. As in the composition was used HPC, why bioadhesion studies were not performed?

The authors appreciate the reviewer offering the reference for the bio-adhesion study. Limited by the instrumentation and materials, the authors realized that the bio-adhesion study hadn’t been performed in the formulation development stages. However, as the authors stated at the end of chapter one, the major purpose of the current investigation is to prove the concept of solvent-free OTF manufacturing and compare the manufactured films across different methods. So, the bio-adhesion study will be performed in future formulation optimization works.

  1. There are many errors. Please check the English and formatting.

Apologies for the discrepancies and typos. The whole manuscript has now been carefully reviewed and all typos have been corrected.

Reviewer 2 Report

Very nice work. The authors have completed a very thorough technical comparison of OTFs prepared by different solvent-free processes. This includes very comprehensive physicochemical characterization and comparison of the various films (quality of amorphous dispersions, texture analysis, mechanical properties). A few suggestions provided to address some of the challenges with potential application.

Author Response

It's great pleasure to gain recognition from the reviewer and thanks for pointing out the issues and misspells. Please find the authors' answers in the attached file. 

Reviewer 3 Report

Dear Authors,

You presented a very nice piece of work. My overall merit is high, but there is one missing thing - units for variables. It is pretty annoying and makes me confused. I strongly suggest providing units for time and calculated constants. That will avoid confusion from readers that will try to repeat calculations.

Author Response

It's great pleasure to gain recognition from the reviewer. The authors checked the units in the manuscript thoroughly and fixed them where appropriate.

Round 2

Reviewer 1 Report

The authors performed the requested revisions. Now it can be accepted for publication.